# The Effect of Statistical Downscaling on the Weighting of Multi-Model Ensembles of Precipitation

**Adrienne M. Wootten** [1,*] **, Elias C. Massoud** [2] **, Agniv Sengupta** [2] **, Duane E. Waliser** [2] **and Huikyo Lee** [2]

1    South Central Climate Adaptation Science Center, University of Oklahoma, Norman, OK 73019, USA
2    Jet Propulsion Laboratory, California Institute of Technology, Pasadena, CA 91109, USA;
     elias.massoud@jpl.nasa.gov (E.C.M.); agniv.sengupta@jpl.nasa.gov (A.S.);
     duane.e.waliser@jpl.nasa.gov (D.E.W.); huikyo.lee@jpl.nasa.gov (H.L.)
*    Correspondence: amwootte@ou.edu

**Abstract:** Recently, assessments of global climate model (GCM) ensembles have transitioned from using unweighted means to weighted means designed to account for skill and interdependence among models. Although ensemble-weighting schemes are typically derived using a GCM ensemble, statistically downscaled projections are used in climate change assessments. This study applies four ensemble-weighting schemes for model averaging to precipitation projections in the south-central United States. The weighting schemes are applied to (1) a 26-member GCM ensemble and (2) those 26 members downscaled using Localized Canonical Analogs (LOCA). This study is distinct from prior research because it compares the interactions of ensemble-weighting schemes with GCMs and statistical downscaling to produce summarized climate projection products. The analysis indicates that statistical downscaling improves the ensemble accuracy (LOCA average root mean square error is 100 mm less than the CMIP5 average root mean square error) and reduces the uncertainty of the projected ensemble-mean change. Furthermore, averaging the LOCA ensemble using Bayesian Model Averaging reduces the uncertainty beyond any other combination of weighting schemes and ensemble (standard deviation of the mean projected change in the domain is reduced by 40–50 mm). The results also indicate that it is inappropriate to assume that a weighting scheme derived from a GCM ensemble matches the same weights derived using a downscaled ensemble.

**Keywords:** climate change; climate modeling; downscaling; ensemble weighting; multi-model averaging; precipitation

## 1. Introduction

Climate modeling is traditionally directed toward research to improve our understanding of the climate system [1] and to develop climate projections that aid climate change decision-making [2]. However, regional and local scales are not well represented by global climate models GCMs, [3]. Downscaling techniques can be used to reduce the biases of GCMs, translate the GCM-simulated response to local scales, and provide added information for decision-making [4,5]. Downscaling techniques include regional climate modeling and statistical downscaling (SD) and bias correction methods. SD methods are both computationally efficient and flexible, which has led to their use for impact assessments, e.g., [6–10], including the National Climate Assessment NCA, [11].

Numerous assessments of GCMs currently exist, e.g., [12–17], and downscaling is being investigated in programs such as the Coordinated Regional Downscaling Experiments CORDEX, e.g., [18]. Recently, there has been a transition from using an unweighted, multi-model ensemble mean to more advanced methods that account for the skill and independence of models to inform

the weighting strategy [15,16,19,20]. This transition has occurred, in part, from the recognition that some models can have more skill for certain variables and regions, but also because the use of models with common code bases can result in a lack of independence between GCMs [15,16,19–22]. In addition, it has been noted that the spread in the weighted multi-model mean can be interpreted more accurately as a measure of uncertainty while the spread of the unweighted average is not a measure of uncertainty [23,24]. Given this knowledge, the Fourth National Climate Assessment (NCA4) was the first of the NCA reports to use an ensemble-weighting scheme [25]. Its ensemble weights were calculated using an ensemble of GCMs, but the ensemble weights were applied to a statistically downscaled ensemble.

One reason to address model or statistical downscaling (SD) method independence with ensemble weighting is the replication of programming code across institutions [22,26]. While individual GCMs share parts of code from various modeling institutions, one can argue that applying an SD method to an ensemble of GCMs has two significant effects. First, it applies a single common code and statistical framework to all GCMs in the ensemble, which can cause the weighting scheme of a post-SD ensemble to force the weights of all members to be equal [24]. Second, SD methods correct GCM biases based on a single common training dataset which often has biases of its own that can influence the projections [27,28]. In essence, all the members of a post-SD ensemble of GCMs share the biases of a single statistical technique and training dataset, which may further reduce ensemble independence. In this study, our results will provide details on how the model independence is sensitive to the SD method and the weighting scheme used for the model averaging.

Sanderson et al. [19] present a strategy to weight a climate model ensemble that considers both the skill in the climatological performance of individual models as well as their interdependency. The purpose of this weighting strategy is to weigh GCMs to account for historical skill and model independence. This strategy has been implemented in several studies so far, e.g., [15,19,20,22] as well as in NCA4 [11]. Eyring et al. [29] thoroughly discuss the state of GCM evaluation and how others have constrained the spread of uncertainty in future projections. They show that there is a need to better understand advanced weighting methods to provide decision-makers a more tightly constrained set of climate projections. Although it is important to consider a range of future projections in risk assessment, e.g., [30], stakeholders searching for projections express a need for assistance to select useful projections and assess uncertainty, e.g., [31]. Therefore, the scientific community needs to apply model-averaging techniques to more judiciously combine multi-model climate projections while considering the interdependency of the GCMs. One technique to do so is the Bayesian Model Averaging (BMA) method [15,16,32]. BMA produces a multi-model average created from optimized model weights such that the BMA-weighted model ensemble average for the historical simulation closely matches the observation and its uncertainty. In this study, we apply both the BMA weighting strategy and the weighting scheme introduced by Sanderson et al. [19] to study the post-SD effect on the weighting of multi-model ensembles of precipitation for the south-central United States.

To our knowledge, the influence of statistical downscaling on weighted-model projections and the independence of the ensemble has not been examined. In addition, BMA has not yet been implemented with a statistically downscaled ensemble, although examples of recent work include the implementation of BMA in a hybrid approach with quantile mapping [24] in the context of global atmospheric rivers [15] and for precipitation over the contiguous United States [16]. This study will focus on three questions. First, what are the sensitivities of the GCM and statistically downscaled ensembles to the various weighting schemes? Second, will a BMA approach together with statistical downscaling increase the historical ensemble accuracy and improve confidence in the ensemble-mean projected change? Finally, is it appropriate to assume that ensemble weighting derived from GCMs will produce similar results when applied to a GCM and statistically downscaled ensemble? In this manuscript, Section 2 overviews the methods and data used for this analysis. Section 3 presents our results and discussion. Section 4 concludes this work and recommends future work to further this study.

## 2. Materials and Methods

### 2.1. Study Domain and Variables

The south-central United States (from about 26° N 108.5° W to 40° N 91° W) has a varied topography with a sharp gradient in mean annual precipitation from the east (humid) to the west (arid). The region includes the Mississippi River Valley and the Ozark Mountains in the east (elevations of 200–800 m), the Rocky Mountains in the west (1500–4400 m), and the Gulf of Mexico in the southeast (near sea level). Precipitation in the southeast portion of the domain can be eight times higher than drier western locations (Figure 1).

Our study domain excludes Mexico and the Gulf of Mexico, as gridded observations are limited over these areas. The precipitation gradient is challenging for GCMs to represent in this region [28]. Precipitation is also more difficult to simulate in climate models and to downscale, e.g., [14,33–36]. The complex topography and climatological differences in precipitation provide an opportunity to assess the weighting schemes and statistical downscaling, helping generalize the results of this study to other regions.

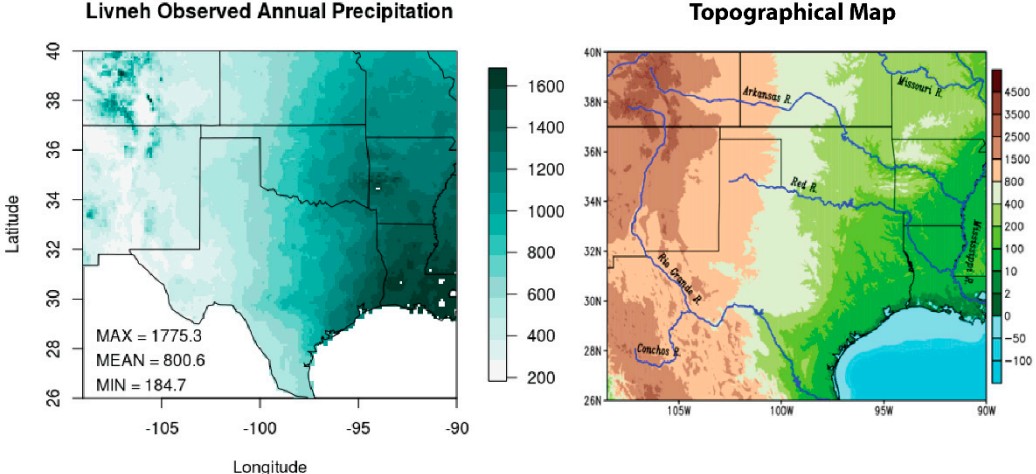

**Figure 1.** Left—Study domain overlaid with annual average precipitation (mm) from Livneh v. 1.2 [37]. Right—Topographical map for the study domain: The elevation map of south-central United States with major rivers overlaid on it. Brown/green shading denotes elevation (in units of m), while the rivers are outlined in blue. Topography, bathymetry and shoreline data are obtained from the National Oceanic and Atmospheric Administration (NOAA) National Geophysical Data Center's ETOPO1 Global Relief Model [38]. This is a 1 arc-minute model of the Earth's surface developed from diverse global and regional digital datasets and then shifted to a common horizontal and vertical datum. River shapefiles are obtained from the Global Runoff Data Centre's Major River Basins of the World [39].

### 2.2. Climate Projection Datasets

We use one member each from 26 GCMs in from the Coupled Model Intercomparison Projection Phase 5 (CMIP5 [40]) archive to form the GCM multi-model ensemble. To form the downscaled ensemble, the same 26 GCMs are used from the downscaled projections created with the Localized Constructed Analogs (LOCA) method [41]. The LOCA-downscaled projections have been used in other studies and the NCA4 [11]. Table S1 lists the GCMs used for both the GCM ensemble (hereafter CMIP5 ensemble) and downscaled ensemble (hereafter LOCA ensemble).

To facilitate analysis, the data for each ensemble member are interpolated from their native resolution to a common 10 km grid across the south-central U.S. using a bi-linear interpolation similar to that described in Wootten et al. [28]. We examine projected precipitation changes from 1981–2005 to 2070–2099 driven by RCP 8.5, which ramps the anthropogenic radiative forcing to 8.5 W/m$^2$ by

2100 [42,43]. We chose RCP 8.5 to maximize the change signals and allow us to analyze greater differences between weight schemes and downscaling techniques. The historical period (1981–2005) is used for both the historical simulations and historical observations (next section) to facilitate comparisons with other efforts in this region [28] and because the historical period of the CMIP5 archive (and SD projections) also ends in 2005 [40]. This historical period is also of similar length to other similar studies, e.g., [13,14,44,45].

### 2.3. Observation Dataset

Although weather stations can be used for training or calibration purposes in statistical downscaling, many publicly available downscaled projections (including LOCA) are created using gridded observation-based data for training. Gridded observations are based largely on station data that are adjusted and interpolated to a grid in a manner that attempts to account for data biases, temporal/spatial incoherence, and missing station data [27,28,46,47]. In this study, we use Livneh version 1.2 (hereafter Livneh [37]) as the gridded observation data used for comparison to the ensembles. The Livneh data served as the training data for the LOCA ensemble, so it is expected that LOCA will be more accurate than the GCMs when compared to the Livneh dataset. The Livneh observations are also interpolated to the same 10 km grid using bilinear interpolation to facilitate analysis. While we recognize that different gridded observations and downscaling techniques influence projections of precipitation, the effect is minimal on mean annual precipitation [28]. Therefore, we hold that it is appropriate to make use of only one SD method and one gridded observation dataset.

### 2.4. Weighting Schemes

In this analysis, we apply an unweighted multi-model mean and four weighting schemes to both the CMIP5 and LOCA ensembles, similar to the experimental design of Massoud et al. [15]. These four weighting schemes are detailed briefly below along with a brief discussion of differences between the Sanderson et al. [20] method and BMA. Figure 2 shows a schematic flowchart of the analysis with the weighting schemes.

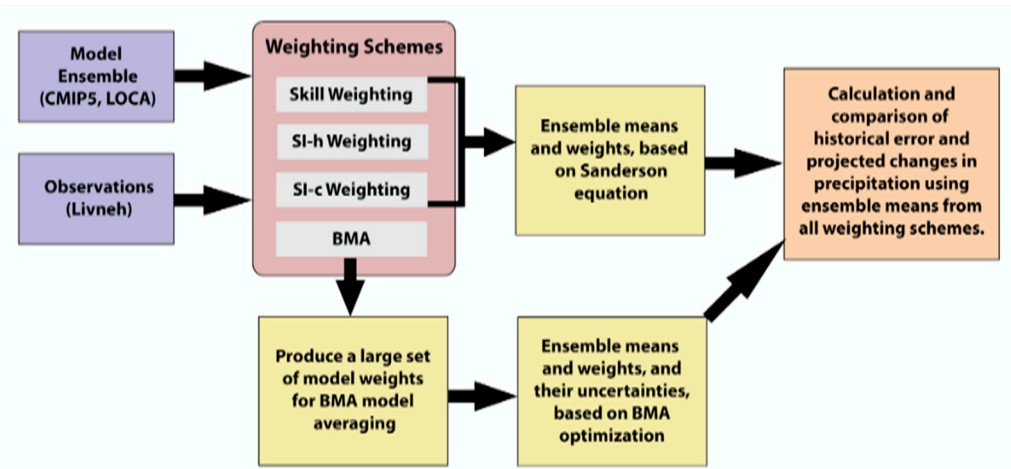

**Figure 2.** Flowchart showing the process of analysis with weighting schemes. The skill weighting uses only the historical skill. The SI-h uses the historical skill and independence is used to determine model weights. The SI-c weighting uses the historical skill and the independence of the projected change signal to determine model weights. Bayesian Model Averaging (BMA) explicitly accounts for both skill and independence during the optimization process. Results from the weighting schemes are compared to the results from the unweighted LOCA and CMIP5 ensembles.

The CMIP5 and LOCA ensembles are supplied to each weighting scheme. All the weighting schemes used in this analysis derive model weights using precipitation in the full domain separately

for both the CMIP5 and LOCA ensembles. The ensemble mean for each combination of weighting scheme and ensemble (CMIP5 and LOCA) is calculated at each grid cell in the domain.

### 2.4.1. Historical Skill Weighting

The first weighting scheme (hereafter Skill) is a simple skill weighting based on the root mean square error (RMSE) of each ensemble member compared to the Livneh gridded observations. The weights are normalized so that the resulting weights of the 26 ensemble members sum to one.

### 2.4.2. Historical Skill and Historical Independence Weighting (SI-h)

The second weighting method is described by Sanderson et al. hereafter SI-h, [20]. The SI-h uses a normalized area-weighted RMSE matrix that compares the RMSE of each model against observations (skill) and each model against all other models (independence). In a multivariate context, a normalized matrix for each variable that is linearly combined using equal weights for each variable. In this study, we use a single normalized RMSE matrix given our focus on precipitation only. A set of skill and independence weights are calculated using the normalized RMSE matrix. For the independence weights, the effective repetition of each model is calculated using the normalized matrix of RMSE. The independence weight of a model is the inverse of the effective repetition. Models that are more alike to other models in the ensemble have a higher (lower) effective repetition (independence weight). The skill weights compare the normalized RMSE of each model against the observations. Models with lower RMSE are given a higher skill weight. Both the independence and skill weights are calculated using a radius of similarity and quality which determines the degree to which a model should be down-weighted for independence and skill. For this analysis we set the radius of similarity and quality to 0.48 and 0.8, respectively, matching Sanderson et al. [20]. The resulting skill and independence weights for the ensemble members are multiplied together to produce the model weights. These weights also are normalized (i.e., final weights sum to one). SI-h weights are calculated only for our study region.

### 2.4.3. Historical Skill and Future Independence Weighting (SI-c)

The third weighting method is a variation of the SI-h method described in Sanderson et al. [20]. One can argue that statistical downscaling improves the skill of the ensemble in the historical period by training with gridded observations, simultaneously nudging every model toward these observations during the historical period (i.e., bias correction). As such, one would expect that the skill (independence) weights of a downscaled ensemble would be high (low). Still, a primary goal of statistical downscaling is to retain the future-change signal of the GCMs in the downscaled output [48]. Therefore, it is appropriate to examine the effect of downscaling on the independence of the change signals in the multi-model ensemble. To do so, we alter SI-h by calculating the normalized area-weighted RMSE matrix using the projected change in precipitation from each model in the ensemble for the independence weighting. That is, the independence weighting in this adjusted scheme (hereafter SI-c) focuses on the *models' effective repetition of the change signal* instead of the *historical climatology*. The SI-c skill weights are identical to those in SI-h, and the skill and independence weights are combined and normalized in the same manner as the SI-h scheme to determine the SI-c final model weights.

### 2.4.4. Bayesian Model Averaging

Hoeting et al. [32] provide a comprehensive overview of Bayesian Model Averaging (BMA) and its variants, which have been widely used in previous literature. BMA differs from other model-averaging methods by explicitly estimating the weights and associated uncertainties of the weights for each model through maximizing a specified likelihood function. In other words, BMA obtains model weights that produce model combinations that have the maximum likelihood of matching historical observations compared to other model combinations. Using these optimized weights, BMA constructs

the mean and uncertainty distribution of the performance metrics (or objective function) of interest. Applications of BMA have been described in numerous works [15,16,49–57].

The BMA method offers an alternative to the selection of a single model from several candidate models, by weighting each candidate model according to its statistical evidence, which is proportional to the model's skill and independence. Since the BMA method estimates a distribution of model weights, various model combinations become possible, which implicitly addresses the model dependence issue. In other words, consider that in the BMA framework there is a hypothetical 'Model A' and a 'Model B' that are similar and therefore not independent; 'Model A' may have higher weights in some combinations, and conversely, 'Model B' might have higher weights in other combinations. Consequently, if both models are rewarded in the same set of weights, each model likely receives a reduced weight because both models are providing information to the model average. Therefore, model dependence can play a role in the BMA scheme since both of the dependent models can affect each other's weights, which can be portrayed in the posterior samples. See Massoud et al. [15] and the supplementary information of Massoud et al. [16] for more information on how BMA is implemented.

Successful use of the BMA/Markov Chain Monte Carlo (MCMC) application depends on many input factors, such as the number of chains, the prior used for the parameters, the number of generations to sample, the convergence criteria, among other things [58]. For our application, we used C = 10 chains, the prior was a uniform distribution from 0–1 for each model weight, each sampled set of weights was normalized so that the sum of weights equaled 1, the number of generations was set at G = 5000 for each metric being fit, and the convergence of the chains relied on the Gelman and Rubin [59] diagnostic, where we applied the commonly used convergence threshold of R = 1.2.

### 2.4.5. Differences between SI-h and BMA

Similar to how other studies have defined dependence [15,19,20], we define it here as the 'distance' of the models in the 'model space'. This definition indicates that a metric like RMSE describes dependence. The closer two models are (i.e., the lower the RMSE between them), the more dependent they are and therefore provide similar information to the model ensemble. Conversely, the larger the RMSE (i.e., the larger the 'distance') between two models, the more independent they are. In the Sanderson approach, shown in Sanderson et al. [19,20] and Massoud et al. [15], there is an explicit consideration of both the skill and the dependence of the model ensemble when calculating the model weights. This explicit consideration only covers the first-order effect, however, by slightly nudging model weights based on a quantitative assessment of model skill and dependence. This quantitative assessment is subjective because the user can choose whether to apply higher weights for dependent models or higher weights for skillful models. In comparison, the BMA method is objective, through the use of the objective function (or likelihood function), and is a method that captures the total order of the model weight combinations. The field of climate research still warrants proper model averaging methods see [29]. We believe that the combination of methods described in our paper (i.e., Sanderson approach and the BMA method) offers a way to infer the total distributions of these model combinations based on statistics that the user desires.

## 3. Results and Discussion

To begin, we explore the influence of statistical downscaling on the ensemble weighting followed by an examination of the historical accuracy and projected changes resulting from each ensemble.

### 3.1. Ensemble Weights

Applying the SI-h and SI-c approaches, one can examine the behavior of the skill and independence weights separately. The SI-h weighted CMIP5 ensemble has both smaller skill weights and greater independence weights than the SI-h weighted LOCA ensemble (Figure 3).

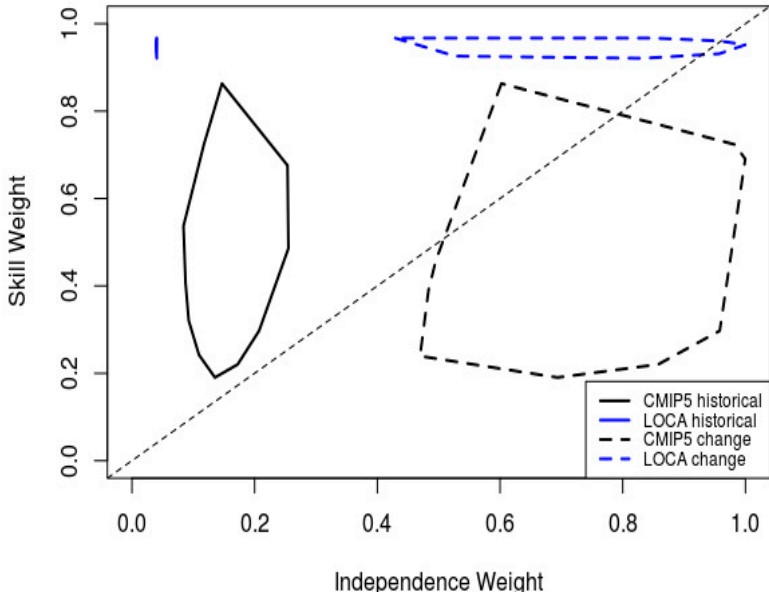

**Figure 3.** Skill and Independence weights for the CMIP5 (black) and LOCA (blue) ensemble. The solid lines are for the ensembles using only historical precipitation information (SI-h). The dashed lines make use of the projected change in precipitation to derive independence weights and historical accuracy to derive skill weights (SI-c). The individual weights for each ensemble member are contained within each polygon.

This result is to be expected given the nature of statistical downscaling with GCMs. Each member of the ensemble downscaled with LOCA is bias-corrected to closely resemble the training data in the historical period. This connection, in turn, makes the LOCA downscaled ensemble more skillful compared to the observations but also makes each ensemble member in the historical period alike to the training data used for downscaling. That is, each downscaled ensemble member now shares in common the errors of the training data used. As a result, the weights from the CMIP5 and LOCA ensembles can be quite different, though the method used to craft the ensemble weights is the same. The average skill weights for the CMIP5 and LOCA ensembles using SI-h are 0.44 and 0.95, respectively. The average independence weights for the CMIP5 and LOCA ensembles using SI-h are 0.13 and 0.040, respectively.

Recall that the SI-c method examines the independence of the ensemble change signal alongside the historical skill to craft the associated ensemble weights. The SI-c weighted ensembles have higher independence weights than the SI-h ensembles, but they retain the same skill weights. As such, the LOCA SI-c ensemble has a higher skill weight than the CMIP5 SI-c ensemble but also retains a similar range of independence weights as compared to the CMIP5 SI-c ensemble. The average independence weight for the LOCA SI-c ensemble (0.69) is slightly lower than that of the CMIP5 SI-c ensemble (0.72). This result suggests that the statistical downscaling technique may have little influence on the independence of the ensemble. In addition, precipitation is one of the most challenging variables to represent in climate modeling, e.g., [34] and this study developed the ensemble weights for each scheme using precipitation only. Using a multivariate approach to developing the ensemble weights may further decrease the independence weights of the downscaled ensemble.

Figure 4 shows the model weights estimated using the various methods. The difference between the SI-h and SI-c approaches to independence is shown by the resulting overall weights of each approach (Figure 4).

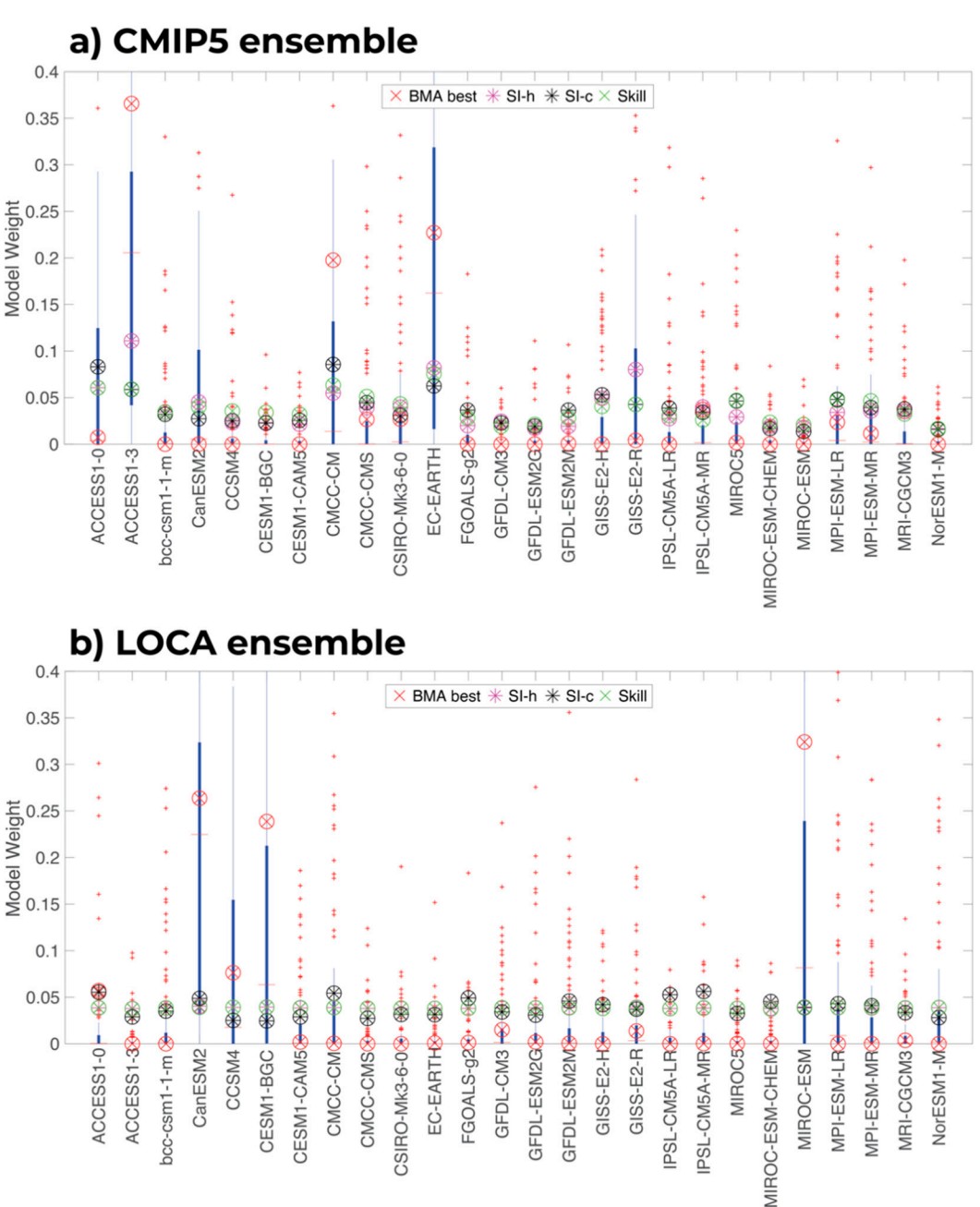

**Figure 4.** (**a**) BMA best, SI-h, SI-c and Skill weights for the CMIP5 ensemble; (**b**) BMA best, SI-h, SI-c and Skill weights for the LOCA ensemble. Boxplots in the background are the weights across all BMA simulations.

The focus only on the historical period causes the SI-h and skill weights to be nearly identical in the LOCA ensemble (where each member has a nearly equivalent weight). However, the SI-c weights are more variable than the SI-h weight and have unique weights for each ensemble member. That said, for both CMIP5 and LOCA ensembles, the weights of the Skill, SI-h, and SI-c approaches generally cluster together (Figure 4).

We then look at the BMA weights for the CMIP5 and LOCA ensembles. The BMA method provides a distribution of weights for each model, and Figure 4 shows this distribution compared to other weighting schemes. For both the CMIP5 and LOCA ensembles, the BMA distributions of weights (shown in the blue box-and-whisker plots in Figure 4) tend to span the values estimated with the Skill, SI-h, and SI-c approaches, but also cover a broader region of the model weight space, since many

model combinations are possible with BMA. Furthermore, the best combination estimated with BMA, marked as 'BMA best' in Figure 4, is significantly different from the other weighting schemes.

While formulated slightly differently, the Skill, SI-h, and SI-c weighting schemes look only at one moment of the distribution, essentially using only one sample to estimate the model weights. In contrast, the BMA approach examines multiple moments of the distribution of the model weights using MCMC sampling. The BMA computes a distribution of weights that gives a similar fit to the observations and results in thousands of samples that can be used to propagate the uncertainty from the multi-model ensemble [15,16]. In addition, the BMA approach rewards skillful models while also penalizing co-dependence, but it does so using a greater number of samples compared to the SI-h and SI-c weightings. As such, it is likely the multiple moments used in BMA drive the weighting for models to be significantly different than the SI-c, SI-h, and skill weights.

In addition, one also can observe that the BMA weights (and their distributions) are not the same as those of the CMIP5 and LOCA downscaled ensembles. The top three models in the CMIP5 ensemble based on the BMA best weights are ACCESS 1-3, EC-EARTH, and CMCC-CM (Figure 4a). The top three models in the LOCA ensemble based on the BMA best weights are CanESM2, CESM1-BGC, and MIROC-ESM (Figure 4b). Statistical downscaling bias corrects each GCM, inherently improving the historical skill. However, given the common modeling framework that statistical downscaling applies to each GCM, there is an additional level of co-dependence applied to each GCM. In the SI-h and SI-c, the resulting pattern of the weights is similar across both ensembles, though the magnitude of the weights' changes. Given that the BMA weights are quite different between ensembles, it is plausible that the BMA weighting trained with the LOCA ensemble responds to and captures the changes to skill and co-dependence in the ensemble more effectively than the SI-c and SI-h.

### 3.2. Historical Biases and RMSE

In the historical period (1981–2005), the variability of the within-ensemble members is different pre and post-SD. The LOCA ensemble members are inherently less variable than those of the CMIP5 ensemble (Figure 5, left) as a result of the bias correction performed during the downscaling process.

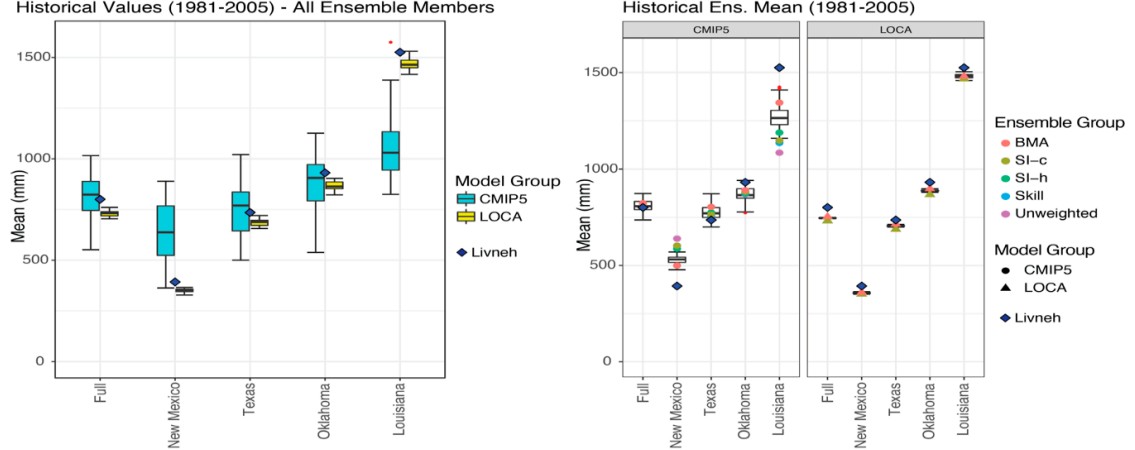

**Figure 5.** Left—Historical values for CMIP5 and LOCA ensembles with the Livneh observed values averaged over the full domain and subdomains. Right—Weighted ensemble mean values with Livneh observed values and boxplot of BMA simulated ensemble means averaged over the full domain and subdomains.

For each ensemble, the variabilities of the BMA-simulated means also are lower than their corresponding raw ensemble (Figure 5, right). This increased confidence in the ensemble mean is reflected in the associated standard deviations (Table 1).

**Table 1.** Standard Deviation (mm) of Historical Ensemble Means—Unweighted CMIP5 and LOCA ensembles vs. weighted CMIP5 and LOCA ensembles for the full domain and all subdomains.

| Group—Weighting | Full | New Mexico | Texas | Oklahoma | Louisiana |
|---|---|---|---|---|---|
| CMIP5—Unweighted | 119.96 | 149.24 | 133.61 | 140.29 | 216.42 |
| CMIP5—Skill | 111.37 | 139.51 | 124.86 | 127.60 | 226.01 |
| CMIP5—SI-h | 122.63 | 139.91 | 138.72 | 137.11 | 255.65 |
| CMIP5—SI-c | 112.91 | 134.55 | 125.47 | 132.19 | 225.47 |
| CMIP5—BMA | 30.62 | 20.96 | 38.51 | 38.38 | 62.52 |
| LOCA—Unweighted | 13.96 | 9.60 | 18.19 | 22.39 | 29.18 |
| LOCA—Skill | 13.94 | 9.59 | 18.16 | 22.36 | 29.15 |
| LOCA—SI-h | 14.00 | 9.62 | 18.23 | 22.42 | 29.28 |
| LOCA—SI-c | 14.14 | 9.99 | 18.36 | 21.90 | 29.74 |
| LOCA—BMA | 1.85 | 2.42 | 3.14 | 3.92 | 11.07 |

In addition, the best BMA weighted ensemble means for the entire domain are consistently closer to Livneh for both the CMIP5 and LOCA ensembles. This result also is reflected in the RMSE of the weighted ensemble means (Figure 6).

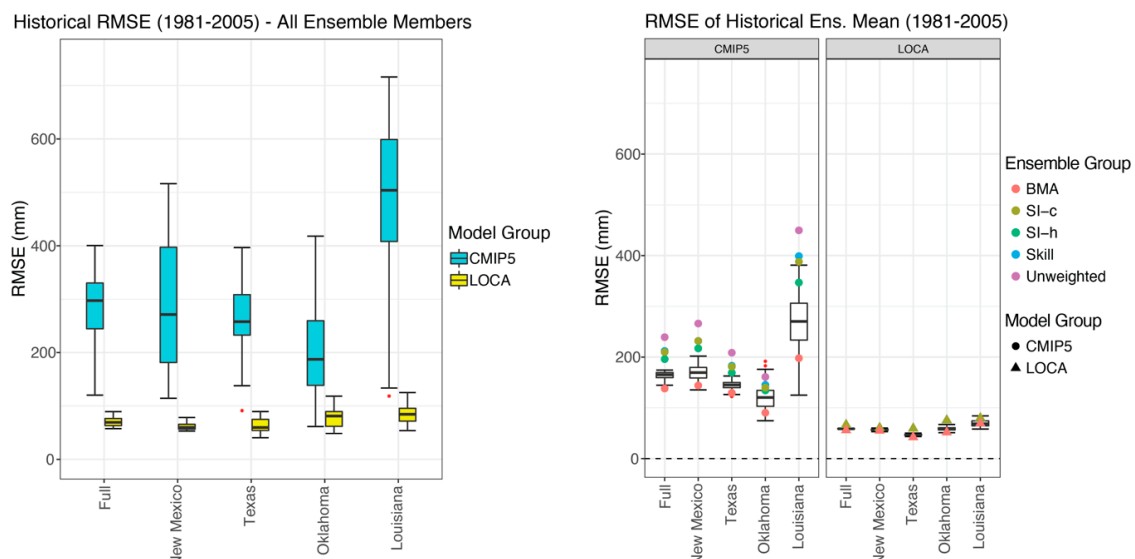

**Figure 6.** Left—RMSE of CMIP5 and LOCA ensembles. Right—RMSE of weighted ensemble means with boxplot of BMA simulated RMSE of ensemble means.

Although LOCA reduces the RMSE of each ensemble member, any weighting scheme tends to reduce the RMSE of the ensemble mean compared to the unweighted mean. In addition, the ensemble mean across the domain created using the best BMA weights has the lowest RMSE compared to the observations.

From Figures 5 and 6, we note that the LOCA ensemble itself is more accurate than the CMIP5 ensemble for precipitation, but also tends to underestimate precipitation in the south-central United States. In addition, we also note that the BMA best weighting does not provide the most accurate ensemble mean of the BMA simulations for individual states (Figure 6). All the weighting schemes used in this analysis derive weights using precipitation in the full domain. The resulting weighting schemes may not produce the most accurate ensemble means in regional subsets. One could repeat the derivation of weights for individual states or smaller regions. Therefore, the results of the analyses presented in this section and the following sections are still valid for this and other regions.

### 3.3. Projected Changes

Figure 7 displays the mean projected change and also highlights unique differences between ensemble means.

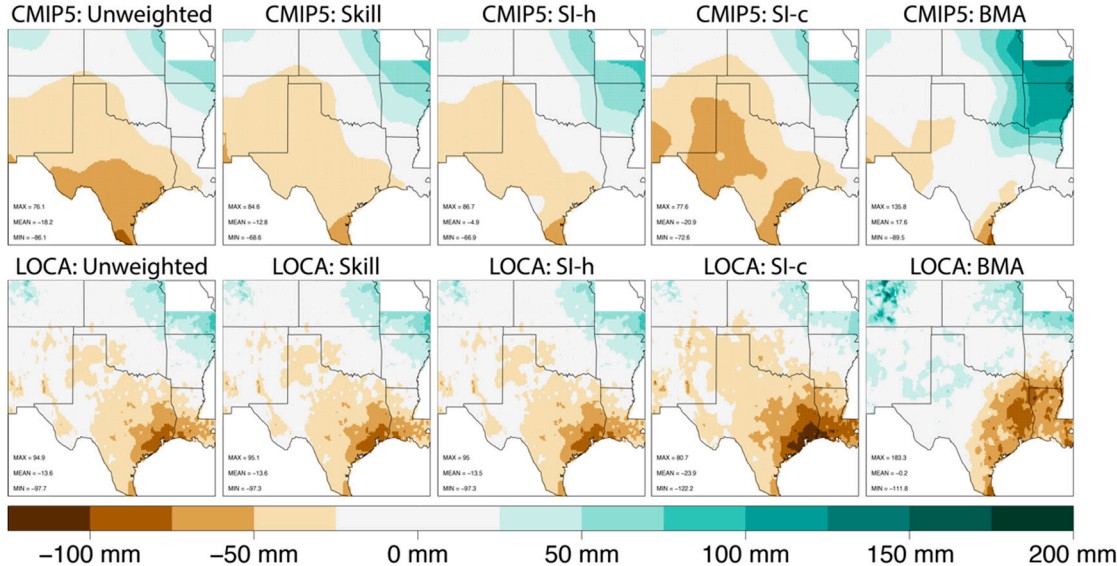

**Figure 7.** Projected change (mm) in annual average precipitation from the unweighted ensembles (left hand column) and all eight weighted ensembles.

Although the spatial pattern for the mean projected change is similar between the unweighted, Skill-weighted, SI-h weighting, and SI-c weighting, the ensemble mean based on the BMA best weighting has a slightly different pattern (Figure 7). For the CMIP5 ensemble, the BMA ensemble mean projects a significant increase in precipitation in the northeast portion of the domain with little to no decrease in the southwest portion of the domain. For the LOCA ensemble, the BMA ensemble mean projects an increase in precipitation across much of the northern portion of the domain, with significant drying in eastern Texas and Louisiana. For the SI-h, SI-c, skill, and unweighted ensembles, the spatial patterns remain consistent across both weighting schemes and ensembles (though the magnitudes are different). Lastly, we notice in Figure 6 that the spatial pattern of the BMA mean projected change is different between the CMIP5 and LOCA ensembles.

The mean projected change across the region from the raw ensembles also is more variable, and thus more uncertain, than the corresponding BMA simulations (Figure 8).

However, the distribution of the mean projected change from the BMA is not identical between the LOCA and CMIP5 ensembles either across the domain or for individual states. In addition, the BMA best weighted mean does not share the same pattern between ensembles while the other weighting schemes demonstrate a similar pattern for mean projected change across the region across ensembles. This result reflects the differences in the weights between ensembles, as BMA accounts for the downscaling procedure. The BMA best weighting generally improves the confidence in the mean projected change over the other weighting schemes or unweighted mean (Table 2). Finally, the combination of the LOCA and BMA results in the smallest uncertainty associated with this region and three of four states, as compared to all other combinations tested in this analysis.

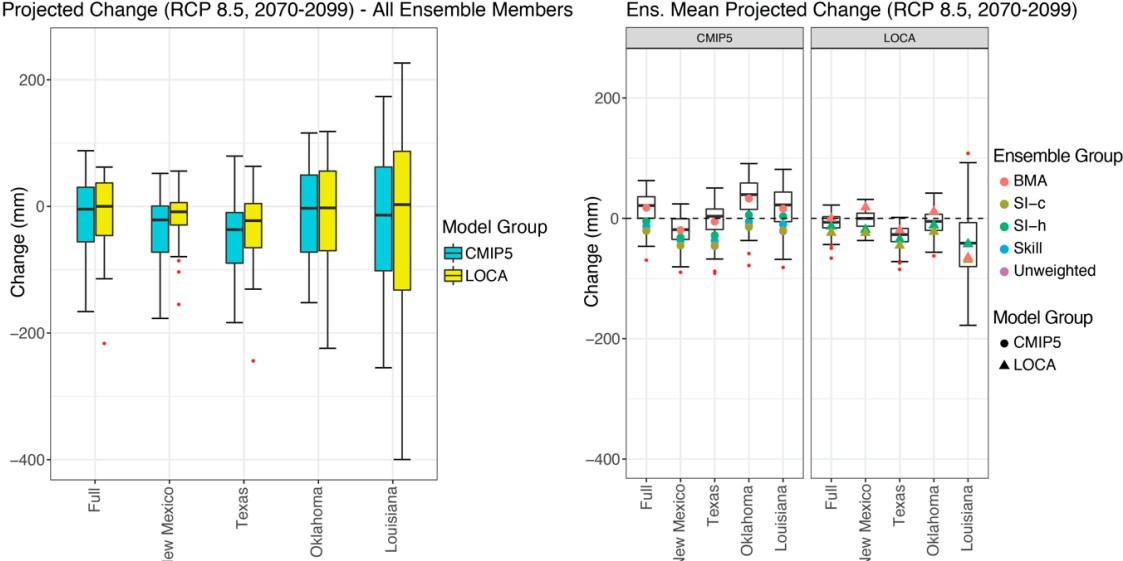

**Figure 8.** Left—Projected change (mm) from CMIP5 and LOCA ensembles. Right—Mean projected change of weighted ensembles with boxplot of BMA simulated mean change.

**Table 2.** Standard Deviation (mm) of Ensemble Mean Projected Change—Unweighted CMIP5 and LOCA ensembles vs. weighted CMIP5 and LOCA ensembles for the full domain and all subdomains.

| Group—Weighting | Full | New Mexico | Texas | Oklahoma | Louisiana |
|---|---|---|---|---|---|
| CMIP5—Unweighted | 62.82 | 60.59 | 69.28 | 77.19 | 116.53 |
| CMIP5—Skill | 64.19 | 59.03 | 73.16 | 81.50 | 111.35 |
| CMIP5—SI-h | 68.19 | 63.40 | 76.26 | 83.84 | 116.78 |
| CMIP5—SI-c | 67.40 | 61.51 | 77.11 | 84.36 | 118.63 |
| CMIP5—BMA | 26.62 | 24.68 | 28.62 | 34.51 | 36.73 |
| LOCA—Unweighted | 65.06 | 46.59 | 69.85 | 82.75 | 178.51 |
| LOCA—Skill | 64.99 | 46.57 | 69.79 | 82.61 | 178.31 |
| LOCA—SI-h | 64.88 | 46.60 | 69.69 | 82.53 | 178.26 |
| LOCA—SI-c | 70.87 | 52.31 | 75.72 | 87.78 | 188.64 |
| LOCA—BMA | 16.32 | 14.87 | 19.05 | 21.55 | 59.39 |

*3.4. Implications of Results*

To the best of our knowledge, this study is the first to comparatively analyze multiple ensemble weighting schemes and the interaction with statistical downscaling. In addition, it appears to be the first study using BMA with a set of statistically downscaled projections to determine ensemble weighting used for both historic and future model simulations of precipitation. In prior efforts to derive ensemble weighting, including the NCA4 [25], the implicit assumption has been that the ensemble weighting derived using a GCM ensemble will match the weights derived using a downscaled ensemble. The previous work of Sanderson et al. [20] shows the common biases and co-dependence associated with shared code between GCMs, and we have extended this logic to the added process of statistical downscaling. To create a statistically downscaled set of projections, a single statistical modeling process is applied to all GCMs used to improve the spatial resolution and correct biases toward a single common set of training data. Therefore, each member of a downscaled ensemble now shares the biases of the statistical downscaling technique and the training data used. That is, using statistical downscaling (applying similar coding to all ensemble members) introduces additional co-dependences into the ensemble in a similar fashion to what is already known for GCMs [20]. This result is confirmed in our paper, as displayed in Figure 3 which shows how the LOCA historical

independence weights are relatively low for all models in the ensemble, compared to those of the historical CMIP5 model ensemble.

While somewhat evident when using the modified Sanderson approach (SI-c), the influence of downscaling on the ensemble is most evident in the BMA results. The BMA is designed to determine the appropriate weighting based on thousands of simulations and the multiple moments of the resulting distribution [15]. BMA rewards the skillful model while penalizing co-dependence. While the Sanderson approach (both SI-h and SI-c) examines only a single moment of the distribution, BMA's incorporation of multiple moments of the distribution allows it to more thoroughly address both concerns of co-dependence and historical skill. As a result, the BMA more capably captures the additional co-dependence and skill introduced by statistically downscaling an ensemble. The consideration of the additional moments of the distribution also allows BMA to reduce the uncertainty of the mean projected changes beyond the Sanderson approaches or skill weighting. These results are also in line with the work of Massoud et al. [15,16].

The BMA generally reduces the uncertainty of the mean projected change of the ensemble beyond any of the other weighting schemes used in this study (which is in line with results from [15] and [16]); it is also apparent that using BMA with the LOCA downscaled projections reduces the uncertainty even further than BMA with the CMIP5 ensemble alone (which is similar to the results suggested in [60] which combined multiple statistically downscaled projections). This result stems from the significant amount of bias correction in the LOCA downscaling method, with the BMA framework further reducing biases from the multi-model ensemble. Given the results of this study, we find that it is not appropriate to assume that the weighting scheme derived from a GCM ensemble would match those derived using a downscaled ensemble (particularly when using BMA). Hence, we recommend the use of multiple weighting methods, if possible, when combining multi-model ensembles to ensure that the model averaging does not solely rely on a few models and to maintain the total order of the distributions of the variables of interest. We also recommend that multiple weighting schemes be considered with downscaled projections for future assessments.

Precipitation remains among the most challenging variables to simulate in global climate models, e.g., [14,33,34,61]. Although this study focused solely on mean annual precipitation, we speculate that our conclusions will remain valid in multivariate analysis incorporating variables that are less challenging to model or thresholds of variables (such as the number of days with rainfall over 1 inch). Such an analysis is planned for our future work. Although this analysis focused on the south-central United States, the various weighting schemes have been used individually in multiple other regions around the world, e.g., [15,16]. Therefore, we also speculate that the results of this analysis can be generalized to other regions. In addition, all of the metrics for the weighting schemes were calculated using the entire south-central U.S., but the uncertainty associated with each state in the region likely can be reduced further by training weighting schemes with a downscaled ensemble for individual states. As such, for individual states or adaptation decision making in smaller regions, it is likely beneficial to consider using weighting schemes tailored for those regions.

## 4. Conclusions

In this study, we have compared four ensemble weighting schemes on mean annual precipitation for both GCM output and the same output after downscaling with a single statistical downscaling method. Traditionally, prior efforts involving ensemble weighting calculate the weights using the raw GCM ensemble and implicitly assume that the same weights apply to a statistically downscaled ensemble (i.e., the weight-transfer assumption). The weight-transfer assumption ignores the additional co-dependence and improved skill introduced by statistical downscaling to all members of the ensemble. The effect of these attributes is most strongly reflected by the BMA weighting scheme, whose weights change dramatically between the CMIP5 and LOCA ensemble. The use of statistical downscaling generally reduces the uncertainty and improves the accuracy associated with the historical mean and mean projected change in annual precipitation (Tables 1 and 2, Figures 5, 6 and 8). However,

the BMA weighted LOCA ensemble also reduces the uncertainty of the mean projected change in annual precipitation above and beyond the other three weighting schemes (Table 2).

While we recommend using multiple weighting schemes, we recognize this may be impractical for large-scale assessments (such as the NCA). However, it may be beneficial to compute the weights for smaller regions when conducting state or local impact assessments or adaptation decisions. Alternatively, one also may consider using a BMA process to select a subset of a multi-model ensemble specific to the needs of stakeholders, for their domain and variables of interest. The development of the associated evaluation and selection framework for these products for smaller domains will greatly benefit local climate change impact assessments and is also the subject of our future work on this topic.

**Supplementary Materials:** The following are available online at http://www.mdpi.com/2225-1154/8/12/138/s1, Table S1: Global Climate Models used to create both the CMIP5 and LOCA ensembles.

**Author Contributions:** Conceptualization, A.M.W.; methodology, A.M.W. and E.C.M.; formal analysis, investigation, A.M.W. and E.C.M.; writing—original draft preparation, A.M.W.; writing—review and editing, A.M.W. and E.C.M.; visualization, A.M.W., E.C.M., and A.S.; supervision, D.E.W. and H.L.; All authors have read and agreed to the published version of the manuscript.

**Funding:** This work was supported in part by National Aeronautics and Space Administration. This material is based upon work supported in part by the National Aeronautics and Space Administration under Grant No. NNX15AK02H issued through the NASA Oklahoma Space Grant Consortium.

**Acknowledgments:** The authors thank the reviewers for their comments and critiques to strengthen this article. A portion of this research was carried out at the Jet Propulsion Laboratory, California Institute of Technology, under a contract with the National Aeronautics and Space Administration. Copyright 2020.

**Conflicts of Interest:** The authors declare no conflict of interest.

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
