# Peer review of "The Effect of Statistical Downscaling on the Weighting of Multi-Model Ensembles of Precipitation"

_climate, doi:10.3390/cli8120138_

Round 1

Reviewer 1 Report

I enjoyed reading this paper and would recommend its acceptance with minor/moderate revisions.

Overall, the paper is well written, its content logically presented, and the main finding supported by data and analyses.

The only two comments and concerns I have are related to completely absent so-called weather generators in the section on literature review. Weather generates play a significant role in downscaling and should be mentioned. Several suggested references are below:

S Fatichi, et al, 2011, Simulation of future climate scenarios with a weather generator, Advances in Water Resources 34 (4), 448-467

X Li, V Babovic, 2019, Multi-site multivariate downscaling of global climate model outputs: an integrated framework combining quantile mapping, stochastic weather generator and Empirical Copula approach, Climate Dynamics 52 (9-10), 5775-5799

Furthermore, bringing precipitation to high spatio-temporal scales is also an outstanding challenge in precipitation simulation. These challenges should also be highlighted in the literature review section. Without these considerations, the paper would be incomplete, and I strongly recommend authors to amend the manuscript accordingly.

A Paschalis et al, 2013, A stochastic model for high‐resolution space‐time precipitation simulation, Water Resources Research 49 (12), 8400-8417

X Li et al, 2018, Three resampling approaches based on method of fragments for daily‐to‐subdaily precipitation disaggregation, International Journal of Climatology 38, e1119-e1138

Reviewer 2 Report

I am willing to recommend the acceptance of this manuscript, pending on some minor reviews. See below.

L14: Add ensembles before have.

L15: Erase ensembles.

L28: Erase sentence starting with We recommend. (Move it to the conclusions.)

L39: Add (SD) after downscaling.

L40: Erase (SD methods).

L68: erase as the and add their; erase of models.

L87: Replace with by in.

L188: Erase (BMA).

L381: Replace that has used with using.

L384: replace demonstrated with showed.

L404: “any other weighting scheme” where? In the literature?

L429: Add this reference after models:

Sempreviva, A. M., M. E. Schiano, S. Pensieri, A. Semedo, R. Tomé, R. Bozzano, M. Borghini, F. Grasso, L. L. Soerensen, J. Teixeira, C.Transerici, 2009: Observed development of the vertical structure of the marine boundary layer during the LASIE experiment in the Ligurian Sea. Annales Geophysicae, 28, 17-25.

Reviewer 3 Report

On the article The Effect of Statistical Downscaling on the Weighting of Multi-Model Ensembles of Precipitation

The authors applied four ensemble-weighting schemes for model averaging to precipitation projections in the south-central United States, and argue that the study is distinct from prior research because it compares the interactions of ensemble-weighting schemes with GCMs and statistical downscaling to produce summarized climate projection products.

The research proposal is interesting and the title is suitable because it reflects the content and emphasize the paper's interest and significance.

Recommendation 1: Abstract

The abstract is good but still need improvement to include quantitative indicator(s) concerning the main article question whish is “What is the effect of statistical downscaling on the weighting of multi-model ensembles of precipitation?” The current version only indicates that the new procedure improved the ensemble accuracy and reduced the uncertainty of projections. Therefore, it is mandatory the write how much were the accuracy augmented, as well the uncertainty reduced.

Precipitation also should be included in key-words.

Recommendation 2: Introduction

This section is very good.

Recommendation 3: Materials and Methods

  1. Material and Methods

2.1. Study Domain and Variables

Figure 1 could include two maps in parallel, (a) the spatial distribution of annual precipitation (already included) and (2) the spatial distribution of terrain elevation (to be included).

2.2. Climate Projections Datasets

Why did you not used the climate normal which means data from 1981 to 2011 to have 30 years of data to perform the simulations? There is not available data from 2006 to present?

(…)

2.4. Methodological Procedure

To be more understandable the authors should draw a detailed flowchart that comprises the entire procedure from data gathering to the final results. The flowchart should be introduced prior to the description of each step of the integrated procedure. Doing so will provide readers with a global vision of what was done, as well as trigger the article clarity.

Examples of flowcharts:

  • Figure 1 of the article https://doi.org/10.1016/j.coastaleng.2018.08.003
  • Figure 3 of the article https://doi.org/10.1016/j.jher.2020.01.006
  • Figure 1 of the article https://doi.org/10.1016/j.oceaneng.2017.12.023
  • Figure 2 of the article https://doi.org/10.1016/j.apenergy.2020.115888
  • Figure 5 of the article http://dx.doi.org/1002/ghg.1982
  • Figure 7 of the article http://dx.doi.org/10.1002/ghg.1944

Recommendation 4: Results

Section 3 (Results) should be changed to Results and Discussion.

The results presentations and discussions must be improved, considering benchmarks with previous works. The current version does not provide discussions of the results. See the example of https://doi.org/10.1016/j.ocemod.2019.101542.

Recommendation 5: Conclusions

The section 4 of the manuscript does not refer to conclusions since it is rather Final Remarks. Even as Final Remarks you should not include speculation in this section (e.g. “… we also speculate that the results of this analysis can be generalized to other regions”). Therefore, I advise you to include some justified speculations, supported by citations that make the statements reasonable, only in Results and Discussions section.

Round 2

Reviewer 3 Report

You have two sections 4 in the manuscript! Choose the one you want to keep in the final version.

Author Response

Thank you so much for your comments! We have corrected this issue and there should now be only one Conclusions section.